

# Gene expression profile of sodium channel subunits in the anterior cingulate cortex during experimental paclitaxel-induced neuropathic pain in mice

Willias Masocha

Department of Pharmacology and Therapeutics, Faculty of Pharmacy, Kuwait University, Safat, Kuwait

Corresponding author
Willias Masocha,
masocha@hsc.edu.kw

## ABSTRACT

Paclitaxel, a chemotherapeutic agent, causes neuropathic pain whose supraspinal pathophysiology is not fully understood. Dysregulation of sodium channel expression, studied mainly in the periphery and spinal cord level, contributes to the pathogenesis of neuropathic pain. We examined gene expression of sodium channel ($Na_v$) subunits by real time polymerase chain reaction (PCR) in the anterior cingulate cortex (ACC) at day 7 post first administration of paclitaxel, when mice had developed paclitaxel-induced thermal hyperalgesia. The ACC was chosen because increased activity in the ACC has been observed during neuropathic pain. In the ACC of vehicle-treated animals the threshold cycle (Ct) values for $Na_v1.4$, $Na_v1.5$, $Na_v1.7$, $Na_v1.8$ and $Na_v1.9$ were above 30 and/or not detectable in some samples. Thus, comparison in mRNA expression between untreated control, vehicle-treated and paclitaxel treated animals was done for $Na_v1.1$, $Na_v1.2$, $Na_v1.3$, $Na_v1.6$, $Na_x$ as well as $Na_v\beta1$–$Na_v\beta4$. There were no differences in the transcript levels of $Na_v1.1$–$Na_v1.3$, $Na_v1.6$, $Na_x$, $Na_v\beta1$–$Na_v\beta3$ between untreated and vehicle-treated mice, however, vehicle treatment increased $Na_v\beta4$ expression. Paclitaxel treatment significantly increased the mRNA expression of $Na_v1.1$, $Na_v1.2$, $Na_v1.6$ and $Na_x$, but not $Na_v1.3$, sodium channel alpha subunits compared to vehicle-treated animals. Treatment with paclitaxel significantly increased the expression of $Na_v\beta1$ and $Na_v\beta3$, but not $Na_v\beta2$ and $Na_v\beta4$, sodium channel beta subunits compared to vehicle-treated animals. These findings suggest that during paclitaxel-induced neuropathic pain (PINP) there is differential upregulation of sodium channels in the ACC, which might contribute to the increased neuronal activity observed in the area during neuropathic pain.

## INTRODUCTION

Voltage-gated sodium channels ($Na_v$) are responsible for action potential initiation and propagation in neurons and other excitable cells. Sodium channels are composed of a pore-forming α subunit associated with one or more auxiliary β subunits that modulate

channel gating, expression and localisation (*Catterall, Goldin & Waxman, 2005*; *Isom, 2001*). There are ten sodium channel α subunits $Na_v1.1$–$Na_v1.9$ and $Na_x$ encoded by genes SCN1A–SCN11A, and four β subunits $Na_v\beta1$–$Na_v\beta4$, encoded by genes SCN1B–SCN4B (*Brackenbury & Isom, 2008*; *Cummins, Sheets & Waxman, 2007*; *Yu & Catterall, 2003*). These sodium channel subunits are expressed in a wide variety of tissues and the level of expression of each channel varies between tissues.

Sodium channels play an important role in the propagation of nociceptive signals. Changes in sodium channel function or expression can result in altered pain sensitivity and perception in various conditions including neuropathic pain (*Bagal et al., 2015*; *Cummins, Sheets & Waxman, 2007*). Dysregulated expression of sodium channels in both the periphery and the central nervous system (CNS), which can result in frequent and ectopic firing in neurons, have been associated with the pathogenesis of neuropathic pain (*Craner et al., 2002*; *Lindia et al., 2005*; *Pertin et al., 2005*; *Rogers et al., 2006*).

In the periphery, the expression all sodium channel α subunits was downregulated, except for $Na_v1.2$, in the dorsal root ganglia (DRG) of rats with spared nerve injury (SNI) (*Laedermann et al., 2014*). Another study observed downregulation of $Na_v1.8$ and $Na_v1.9$ in the DRG of a chronic constriction injury (CCI) model of neuropathic pain (*Dib-Hajj et al., 1999*). However, other studies have observed upregulation of sodium channel subunits such as $Na_v1.3$, $Na_v1.6$, $Na_v1.9$, $Na_v\beta2$ and $Na_v\beta3$ in the DRG of animal models of neuropathic pain (*Craner et al., 2002*; *Lindia et al., 2005*; *Pertin et al., 2005*; *Shah et al., 2001*; *Shah et al., 2000*).

In the spinal cord $Na_v1.3$ was also found to be upregulated in the dorsal horn neurons of CCI and spinal cord injury (SCI) models of neuropathic pain (*Hains et al., 2003*; *Hains et al., 2004*). Sciatic nerve injury (axotomy) resulted in upregulation of $Na_v1.7$ in the spinal cord, which had strong correlation with the level of pain behaviour (*Persson et al., 2009*). In a model of painful diabetic neuropathy there was upregulation of $Na_v\beta3$ expression in spinal cord (*Shah et al., 2001*). $Na_v\beta1$ expression increased whereas $Na_v\beta2$ decreased in the spinal cord of neuropathic rats (*Blackburn-Munro & Fleetwood-Walker, 1999*).

In the brain dysregulation of sodium channel expression has been observed in different areas during neuropathic pain. In the prefrontal cortex $Na_v1.1$ expression was upregulated in mice with SNI (*Alvarado et al., 2013*). The expression of $Na_v1.3$ was upregulated in the ventral posterolateral (VPL) nucleus of the thalamus of rats with CCI or spinal cord contusion injury (*Hains, Saab & Waxman, 2005*; *Zhao, Waxman & Hains, 2006*).

Recently, we observed increased excitability of the anterior cingulate cortex (ACC) to electrophysiological stimulation in a rat model of paclitaxel-induced neuropathic pain (PINP) (*Nashawi et al., 2016*). Paclitaxel is a chemotherapeutic drug whose therapeutic use is sometimes limited by the development of dose-dependent painful neuropathy (*Scripture, Figg & Sparreboom, 2006*; *Wolf et al., 2008*). The ACC is an area in the brain involved in pain perception and modulation, and has increased activity during neuropathic pain (*Hsieh et al., 1995*; *Vogt, 2005*; *Xie, Huo & Tang, 2009*; *Zhuo, 2008*). In previous studies, we observed changes in the expression of gamma-aminobutyric acid

(GABA)-ergic and glutamatergic molecules in the ACC of a mouse model of PINP (*Masocha, 2015a*; *Masocha, 2015b*). However, the expression of sodium channels in the ACC during PINP has not been studied as yet. Studying the expression of sodium channels in the ACC during PINP is important as they might contribute to the increased neuronal excitability, which we observed in the ACC during PINP (*Nashawi et al., 2016*). Thus, in the current study the gene expression of sodium channel subunits in the ACC was evaluated in mice at a time point when the mice had paclitaxel-induced thermal hyperalgesia (*Masocha, 2015a*; *Nieto et al., 2008*; *Parvathy & Masocha, 2013*). In previous studies, gene expression changes of other molecules were observed in the ACC of mice with paclitaxel-induced thermal hyperalgesia (*Masocha, 2015a*; *Masocha, 2015b*).

## MATERIALS AND METHODS

### Animals

Female BALB/c mice (8–12 weeks old; 20–30 g; n = 49) supplied by the Animal Resources Centre (ARC) at the Health Sciences Center (HSC), Kuwait University were used. The animals were kept in temperature controlled (24 ± 1 °C) rooms with food and water given ad libitum. Animals were handled in compliance with the Kuwait University, HSC, ARC guidelines and in compliance with Directive 2010/63/EU of the European Parliament and of the Council on the protection of animals used for scientific purposes. All animal experiments were approved by the Ethical Committee for the use of Laboratory Animals in Teaching and in Research, HSC, Kuwait University.

### Paclitaxel administration

Paclitaxel (Cat. No. 1097, Tocris, Bristol, UK) was dissolved in a solution made up of 50% Cremophor EL and 50% absolute ethanol to a concentration of 6 mg/ml and then diluted in normal saline (NaCl 0.9%), to a final concentration of 0.2 mg/ml just before administration. Mice were treated intraperitoneally (i.p.) for five consecutive days with paclitaxel 2 mg/kg, the cumulative dose was 10 mg/kg, or its vehicle. This treatment regimen produces painful neuropathy and thermal hyperalgesia in mice on day 7 post first administration (*Nieto et al., 2008*; *Parvathy & Masocha, 2013*). A group of control mice was left untreated.

### Tissue preparation and real time RT-PCR

Mice were anesthetized with isoflurane, sacrificed by decapitation on day 7 post first administration of paclitaxel. The ACC was dissected and prepared for RNA extraction as described previously (*Masocha, 2015b*).

Gene transcripts of the 10 sodium channel alpha subunits ($Na_v1.1$, $Na_v1.2$, $Na_v1.3$, $Na_v1.4$, $Na_v1.5$, $Na_v1.6$, $Na_v1.7$, $Na_v1.8$, $Na_v1.9$ and $Na_x$) and four sodium channel beta subunits ($Na_v\beta1$, $Na_v\beta2$, $Na_v\beta3$ and $Na_v\beta4$) were quantified in the ACC of untreated, vehicle-treated and paclitaxel-treated mice by real time polymerase chain reaction (PCR). Total RNA was xtracted from the fresh frozen ACC using the RNeasy Kit (Qiagen GmbH), reverse-transcribed, and the mRNA levels were quantified on an ABI Prism® 7500 sequence detection system (Applied Biosystems) as previously

**Table 1  PCR primer sequences of cyclophilin, and sodium channel subunits.**

| Gene | Polarity | |
|---|---|---|
| | Sense<br>Sequence 5′ to 3′ | Anti-sense<br>Sequence 5′ to 3′ |
| Cyclophilin | GCTTTTCGCCGCTTGCT | CTCGTCATCGGCCGTGAT |
| $Na_v1.1$ | AACAAGCTTCATTCACATACAATAAG | AGGAGGGCGGACAAGCTG |
| $Na_v1.2$ | GGGAACGCCCATCAAAGAAG | ACGCTATCGTAGGAAGGTGG |
| $Na_v1.3$ | GGGTGTTGGGTGAGAGTGGAG | AATGTAGTAGTGATGGGCTGATAAGAG |
| $Na_v1.4$ | CGCGCTGTTCAGCATGTT | CTCCACGTCCTTGGACCAAG |
| $Na_v1.5$ | AGACTTCCCTCCATCTCCAGATA | TGTCACCTCCAGAGCTAGGAAG |
| $Na_v1.6$ | AGCAAAGACAAACTGGACGATACC | CACTTGAACCTCTGGACACAACC |
| $Na_v1.7$ | TCCTTTATTCATAATCCCAGCCTCAC | GATCGGTTCCGTCTCTCTTTGC |
| $Na_v1.8$ | ACCGACAATCAGAGCGAGGAG | ACAGACTAGAAATGGACAGAATCACC |
| $Na_v1.9$ | TGAGGCAACACTACTTCACCAATG | AGCCAGAAACCAAGGTACTAATGATG |
| $Na_x$ | TGTCTCCTCTAAACTCCCTCAG | TGCGTAAATCCCAAGCAAAGT |
| $Na_v\beta1$ | GTGTATCTCCTGTAAGCGTCGTAG | ATTCTCATAGCGTAGGATCTTGACAA |
| $Na_v\beta2$ | GGCCACGGCAAGATTTACCT | CACCAAGATGACCACAGCCA |
| $Na_v\beta3$ | ACTGAAGAGGCGGGAGAAGAC | GGTGAGGAAGACCAGGAGGATG |
| $Na_v\beta4$ | CCCTTGGTGTAGAAACTAAGCAGAG | CAGAAGCGAGTCAGTCAGATACG |

described (*Masocha, 2009*; *Masocha, 2015a*). The primer sequences which were used, listed in Table 1, were ordered from Invitrogen (Life Technologies) and/or synthesized at the Research Core Facility (RCF), HSC, Kuwait University. Threshold cycle (Ct) values for all cDNA samples were obtained and the amount of mRNA of individual animal sample (n = 8–12 per group) was normalized to cyclophilin (housekeeping gene) ($\Delta$Ct). The relative amount of target gene transcripts was calculated using the $2^{-\Delta\Delta Ct}$ method as described previously (*Livak & Schmittgen, 2001*). These values were then used to calculate the mean and standard error of the relative expression of the target gene mRNA in the ACC of paclitaxel- and vehicle-treated mice.

## Statistical analyses

Statistical analyses were performed using Mann Whitney U test using Graph Pad Prism software (version 5.0). The differences were considered significant at $p < 0.05$. The results in the text and figures are expressed as the means $\pm$ S.E.M.

## RESULTS

The mRNA expression of sodium channel subunits were analysed in the ACC at day 7, a time when the mice treated with paclitaxel had developed thermal hyperalgesia as we described previously (*Masocha, 2014*; *Parvathy & Masocha, 2013*) i.e. reduction in reaction latency compared to the baseline latency and vehicle-treated mice (5.7 $\pm$ 0.3 s compared to 9.6 $\pm$ 0.3 and 9.3 $\pm$ 0.3 s, respectively; n = 8 vehicle-treated mice and 10 paclitaxel treated-mice; $p < 0.01$ for both comparisons).

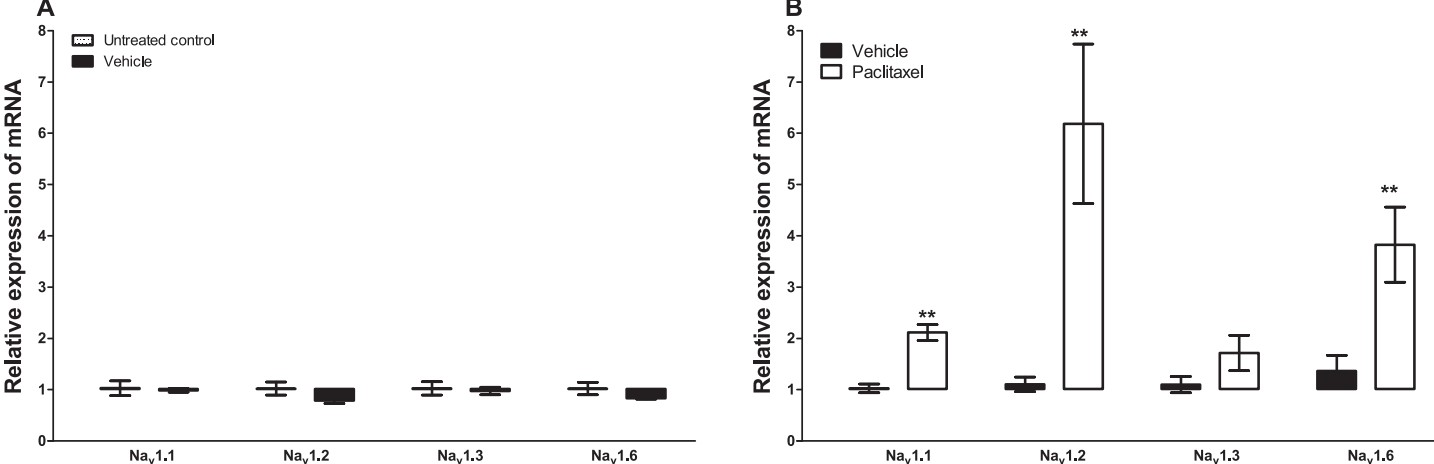

**Figure 1 Effects of paclitaxel on sodium channel alpha subunits transcript levels in the anterior cingulate cortex (ACC).** Relative mRNA expression of sodium channel alpha subunits $Na_v1.1$, $Na_v1.2$, $Na_v1.3$ and $Na_v1.6$ in the ACC of BALB/c mice (A) vehicle-treated mice versus untreated mice. Each bar represents the mean ± S.E.M of the values obtained from four untreated mice and four vehicle-treated mice. (B) Relative mRNA expression of sodium channel alpha subunits on day 7 after first administration of the drug or its vehicle. Each bar represents the mean ± S.E.M of the values obtained from 9 to 11 vehicle-treated mice and 12 paclitaxel-treated mice. ** $p < 0.01$ compared to vehicle-treated mice.

## Expression of sodium channel alpha subunits transcripts in the ACC at seven days after paclitaxel administration

In vehicle-treated animals the Ct values for $Na_v1.4$, $Na_v1.5$, $Na_v1.7$, $Na_v1.8$ and $Na_v1.9$ were above 30 and not detectable in some samples, whereas the Ct values for $Na_v1.1$, $Na_v1.2$, $Na_v1.3$, $Na_v1.6$ and $Na_x$ were below 30. Thus, comparison in mRNA expression between control and paclitaxel treated animals was done for $Na_v1.1$, $Na_v1.2$, $Na_v1.3$, $Na_v1.6$ and $Na_x$.

Treatment with vehicle did not alter the expression of the five sodium channel alpha subunits evaluated, $Na_v1.1$ ($p = 1.000$), $Na_v1.2$ ($p = 0.1143$), $Na_v1.3$ ($p = 0.6857$), $Na_v1.6$ ($p = 0.3429$) and $Na_x$ ($p = 0.3429$), compared to untreated control (Figs. 1A and 2A). Amongst the five sodium channel alpha subunits ($Na_v1.1$, $Na_v1.2$, $Na_v1.3$, $Na_v1.6$ and $Na_x$) treatment with paclitaxel did not significantly alter the mRNA expression of the $Na_v1.3$ ($p = 0.1379$), but significantly increased the expression of $Na_v1.1$ by 2.1 ± 0.2 fold ($p = 0.0002$), $Na_v1.2$ by 6.2 ± 1.6 fold ($p = 0.0003$), $Na_v1.6$ by 3.8 ± 0.7 fold ($p = 0.0051$), compared to vehicle-treated controls (Fig. 1B). $Na_x$ was significantly upregulated by 7.6 ± 2.2 fold ($p = 0.0012$) in the ACC by treatment with paclitaxel compared to treatment with vehicle (Fig. 2B). The most upregulated sodium channel alpha subunits were $Na_v1.2$ and $Na_x$, which were increased by more than sixfold after treatment with paclitaxel.

## Expression of sodium channel beta subunits transcripts in the ACC at seven days after paclitaxel administration

Treatment with vehicle did not alter the expression of three sodium channel beta subunits, $Na_v\beta1$ ($p = 0.2000$), $Na_v\beta2$ ($p = 0.4857$), $Na_v\beta3$ ($p = 0.6857$), but significantly increased the expression of $Na_v\beta4$ ($p = 0.0286$), compared to untreated control (Fig. 3A).
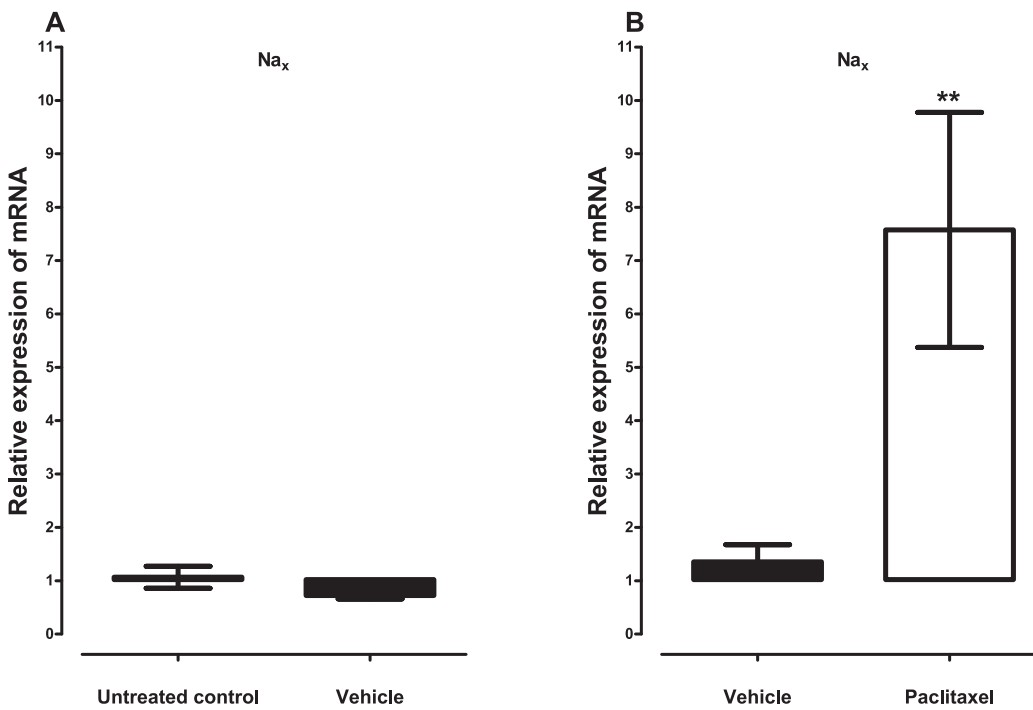

**Figure 2 Effects of paclitaxel on the sodium channel alpha subunit $Na_x$ transcript levels in the anterior cingulate cortex (ACC).** Relative mRNA expression of $Na_x$ in the ACC of BALB/c mice (A) vehicle-treated mice versus untreated mice. Each bar represents the mean ± S.E.M of the values obtained from four untreated mice and four vehicle-treated mice. (B) Relative mRNA expression of sodium channel alpha subunits on day 7 after first administration of the drug or its vehicle. Each bar represents the mean ± S.E.M of the values obtained from 11 vehicle-treated mice and 12 paclitaxel-treated mice. ** $p < 0.01$ compared to vehicle-treated control mice.

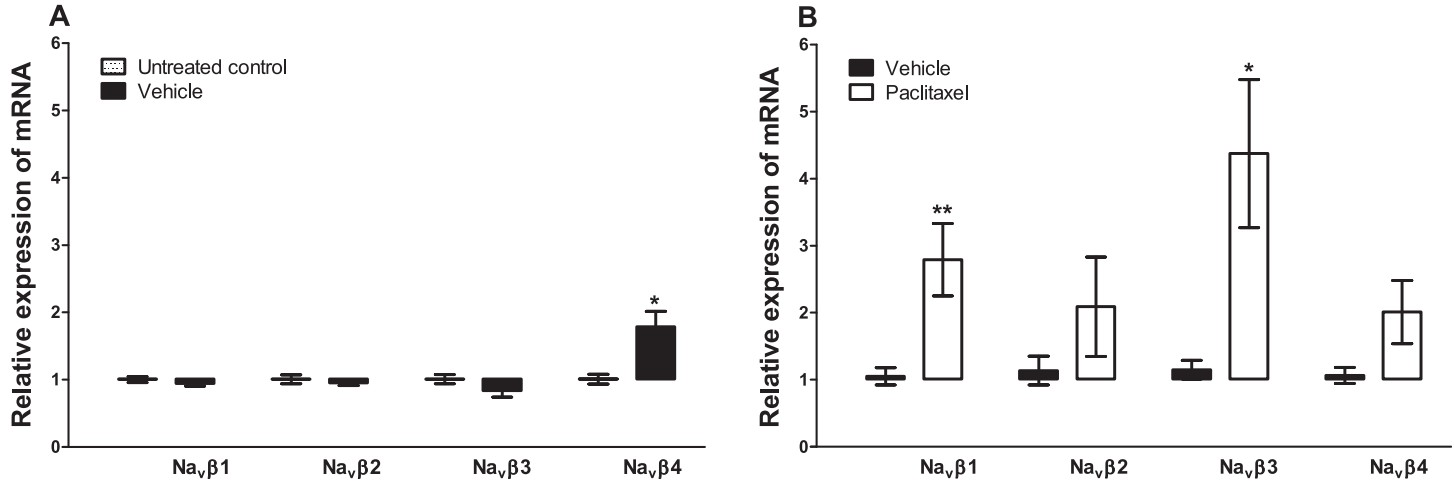

**Figure 3 Effects of paclitaxel on sodium channel beta subunits transcript levels in the anterior cingulate cortex (ACC).** Relative mRNA expression of sodium channel beta subunits $Na_v\beta 1$ to four in the ACC of BALB/c mice (A) vehicle-treated mice versus untreated mice. Each bar represents the mean ± S.E.M of the values obtained from four untreated mice and four vehicle-treated mice. * $p < 0.05$ compared to untreated mice. (B) Relative mRNA expression of sodium channel beta subunits on day 7 after first administration of the drug or its vehicle. Each bar represents the mean ± S.E.M of the values obtained from 8 to 11 vehicle-treated control mice and 8–12 paclitaxel-treated mice. * $p < 0.05$ and ** $p < 0.01$ compared to vehicle-treated mice.

Amongst the four sodium channel beta subunits analysed treatment with paclitaxel significantly increased the expression of Na$_v\beta$1 by 2.8 ± 0.5 fold (p = 0.0047) and Na$_v\beta$3 by 4.4 ± 1.1 fold (p = 0.0127), but not Na$_v\beta$2 (p = 0.2301) and Na$_v\beta$4 (p = 0.0525), compared to vehicle-treated controls (Fig. 3). The most upregulated sodium channel beta subunit was Na$_v\beta$3, which was increased by more than fourfold after treatment with paclitaxel.

## DISCUSSION

This study presents the first comprehensive analysis of the expression of transcripts of sodium channel subunits in the ACC during neuropathic pain, specifically PINP. The ACC is an area of the brain associated with pain perception and modulation (*Vogt, 2005*; *Xie, Huo & Tang, 2009*; *Zhuo, 2008*).

No reports about the expression of sodium channels in the ACC specifically were found. However, Na$_v$1.1, Na$_v$1.2, Na$_v$1.3, Na$_v$1.6 and also Na$_x$ have been reported to be expressed predominantly (but not exclusively) in the brain with differential expression in different brain areas such as hippocampus, thalamus, cerebellum etc. (*Beckh et al., 1989*; *Catterall, 2000*; *Gautron et al., 1992*; *Levy-Mozziconacci et al., 1998*; *Schaller & Caldwell, 2003*; *Westenbroek, Merrick & Catterall, 1989*; *Whitaker et al., 2000*; *Whitaker et al., 2001*). On the other hand, Na$_v$1.4 is expressed principally in the skeletal muscle, Na$_v$1.5 is mainly expressed in cardiac muscle, while Na$_v$1.7, Na$_v$1.8 and Na$_v$1.9 are expressed preferentially in peripheral neurons (*Cummins, Sheets & Waxman, 2007*; *Dib-Hajj, Black & Waxman, 2015*). In the current study using real time PCR all the 10 α subunits and four β subunits were detected in the ACC with different degrees of expression. Na$_v$1.1, Na$_v$1.2, Na$_v$1.3, Na$_v$1.6 and Na$_x$ as well as Na$_v\beta$1–Na$_v\beta$4 were highly expressed in the ACC. On the other hand, although Na$_v$1.4, Na$_v$1.5, Na$_v$1.7, Na$_v$1.8 and Na$_v$1.9 were detected in the ACC they were lowly expressed and/or were not detectable in some samples. Thus, the findings of this study are in agreement with studies described above. This suggests that the different sodium channel subunits have different roles in the ACC and the brain in general. Na$_v$1.1, Na$_v$1.2, Na$_v$1.3, Na$_v$1.6 and Na$_x$ as well as Na$_v\beta$1–Na$_v\beta$4 most likely have more important roles in neuronal activity in the ACC than Na$_v$1.4, Na$_v$1.5, Na$_v$1.7, Na$_v$1.8 and Na$_v$1.9. This could be important for drug development of specific sodium channel blockers; for example a specific blocker of Na$_v$1.1 or Na$_v$1.2 would more likely have more effect in the ACC compared to a specific inhibitor of Na$_v$1.7 or Na$_v$1.8 based on their expression patterns. Further studies are necessary to understand the specific properties and activities of specific sodium channel subunits in the ACC under normal conditions and during neuropathic pain.

Administration of tetrodotoxin (TTX), a voltage-gated sodium channel blocker, was reported to prevent and treat signs of PINP such as thermal hyperalgesia, cold and mechanical allodynia in mice, suggesting that TTX-sensitive voltage-gated sodium channels play a role in the pathophysiology of PINP (*Nieto et al., 2008*). Mexiletine, a non-selective voltage-gated sodium channel blocker was also found to have antinociceptive effects in rats with paclitaxel-induced mechanical allodynia and hyperalgesia (*Xiao, Naso & Bennett, 2008*). However, we found no studies that investigated

the expression of sodium channels in the periphery or CNS during PINP. In the current study, $Na_v1.1$, $Na_v1.2$, $Na_v1.6$ and $Na_x$ as well as $Na_vB1$ and $Na_vB3$ were upregulated in the ACC of mice with paclitaxel-induced thermal hyperalgesia. Upregulation of sodium channel expression has been observed in other areas of the brain during neuropathic pain. In the prefrontal cortex $Na_v1.1$ expression was upregulated in mice with SNI (*Alvarado et al., 2013*). Thus, our data are in agreement with the findings of *Alvarado et al. (2013)* and the suggestion that over-expression of $Na_v1.1$ is involved in increased cortical excitability associated with chronic pain. It is also possible that the increased expression of $Na_v1.2$, $Na_v1.6$, $Na_x$, $Na_vB1$ and $Na_vB3$ in the ACC are involved in the increased excitability of this area observed during PINP (*Nashawi et al., 2016*). Although $Na_v1.3$ was not significantly altered in the ACC during PINP it was reported to be upregulated in the VPL nucleus of the thalamus of rats with CCI and spinal cord contusion injury (*Hains, Saab & Waxman, 2005*; *Zhao, Waxman & Hains, 2006*). The findings of the current study suggest that upregulation of specific sodium channel subunits might contribute to hyperexcitability in the ACC. Hyperexcitability has been associated with dysregulation in sodium channels (*Devor, 2006*). A link between upregulation of $Na_v1.3$ and hyperexcitability of neurons in the spinal cord was found in neuropathic pain after SCI (*Hains et al., 2003*). Recently, we observed increased excitability of the ACC to electrophysiological stimulation in a rat model PINP (*Nashawi et al., 2016*), which could be in part be due upregulation of sodium channels amongst other mechanisms such as decreased GABA availability at the synapse because of increased GABA transporter 1 (GAT-1) expression (*Masocha, 2015b*). Changes in the expression of other molecules such as those of the GABAergic, glutamatergic, muscarinic dopaminergic systems have also been observed in the ACC during experimental neuropathic pain (*Masocha, 2015a*; *Masocha, 2015b*; *Ortega-Legaspi et al., 2011*; *Ortega-Legaspi et al., 2010*). These findings suggest that the ACC plays an important role in the pathophysiology of PINP in addition to other brain areas, the spinal cord and peripheral nerve damage. Paclitaxel has limited ability to cross the blood-brain barrier (*Glantz et al., 1995*; *Kemper et al., 2003*), thus a direct effect of paclitaxel in the ACC is unlikely. In a rat model paclitaxel induced microglial activation in the spinal cord (*Peters et al., 2007*). They proposed (*Peters et al., 2007*) that paclitaxel-induced nerve injury possibly induced neurochemical reorganization within the spinal cord leading to central sensitization (*Cata et al., 2006*) and that the microglial reaction they observed occurred as a result of degeneration of central terminals of injured primary afferent fibers or possibly due to the spinal release of factors from injured neurons rather than direct injury of spinal cord neurons by paclitaxel. In the periphery, paclitaxel causes nerve damage by direct effects on the neurons (*Cavaletti et al., 2000*; *Scuteri et al., 2006*; *Theiss & Meller, 2000*) or via inflammation and the increased infiltration of macrophages into the DRG (*Peters et al., 2007*; *Zhang et al., 2016*), which cause further nerve damage. Thus, the changes observed in the ACC could be due to an increased nociceptive input from the peripheral nerves damaged by paclitaxel resulting in central sensitization. However, information on protein expression is critical to subsequently define the meaning of expression changes in the mRNA level observed in the ACC.

## CONCLUSIONS

In conclusion, the findings of this study show that sodium channel subunit transcripts are differentially expressed in the ACC; with those known to be preferentially expressed in the CNS being highly expressed in the ACC, whereas those known to be preferentially expressed in the periphery being lowly expressed in the ACC. More importantly, the results show that during experimental PINP there is increased expression of various sodium channel subunit transcripts in the ACC, which could contribute to the increased excitability and activity observed in this brain region during neuropathic pain.

## ACKNOWLEDGEMENTS

I am grateful to Dr. Subramanian S. Parvathy, Ms. Salini Soman, Ms. Amal Thomas from the Department of Pharmacology and Therapeutics, Faculty of Pharmacy, for their technical assistance and to the staff from the Animal Resources Centre, HSC, Kuwait University for their support.

### Funding

Funding was provided by Kuwait University Research Sector: PT01/09, SRUL02/13. The funders had no role in study design, data collection and analysis, decision to publish, or preparation of the manuscript.

### Grant Disclosures

The following grant information was disclosed by the authors:
Kuwait University Research Sector: PT01/09, SRUL02/13.

### Competing Interests

The author declares that he has no competing interests.

### Author Contributions

- Willias Masocha conceived and designed the experiments, performed the experiments, analyzed the data, contributed reagents/materials/analysis tools, wrote the paper, prepared figures and/or tables, reviewed drafts of the paper.

### Animal Ethics

The following information was supplied relating to ethical approvals (i.e., approving body and any reference numbers):

All animal experiments were approved by the Ethical Committee for the use of Laboratory Animals in Teaching and in Research, HSC, Kuwait University.

### Data Deposition

The raw data has been supplied as Supplemental Dataset Files.

## Supplemental Information

Supplemental information for this article can be found online at http://dx.doi.org/10.7717/peerj.2702#supplemental-information.

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
