# Peer review of "Gene expression profile of sodium channel subunits in the anterior cingulate cortex during experimental paclitaxel-induced neuropathic pain in mice"

_PeerJ, doi:10.7717/peerj.2702_

## Round 0.1 · original submission · Major Revisions

· Academic Editor

Major Revisions

We are sorry we cannot be more positive at this stage, but according to the reviewers´comments, a major revision of the paper needs to take place before it can be acceptable for publication in PeerJ. If, however, you think you can address all the reviewers´ concerns (especially reviewer#2), we would consider a revised version of the manuscript.

Reviewer 1 ·

Basic reporting

In this manuscript W. Masocha presents an analysis of changes in the expression of the most representative transcripts subunits that compose the voltage-dependent Na+ channels (Nav) in the anterior cingulate cortex (ACC), in the central nervous system, and during the generation of a model of pain caused by paclitaxel administration, a chemotherapeutic drug that apparently presents as a side effect generation of neuropathic pain (phenomenon called PINP along the manuscript). Because the nature of this side effect is still unknown, Masocha studied the effects of paclitaxel treatment on Nav´s subunit expression, molecules responsible for nerve impulse generation and transmission in the central nervous system, and especially their expression in the ACC nucleus, that has shown increased electrical activity during the generation of PINP in previous unpublished work by the author and others.

Experimental design

The approach is original and the question is important to understand the mechanisms involved in pain in this model. The author shows that α subunit transcripts for the Nav1.1, Nav1.2, Nav1.6, Nax, as well as for Navβ1 and Navβ3, all increased their expression while others remain the same or are below the detection level by the method used. It is suggested that this mRNA increase contributes to the concomitant increase in neural activity observed during PINP. The study was conducted rigorously and the methods are well described.

Validity of the findings

The current version of the manuscript required to add the following aspects:
1) Protein expression analysis, mainly for α subunits Nav1.2, Nav1.6, and Navβ3 should be added in this study. Changes in mRNA level may not reflect changes in protein synthesis. Consider also to add evidence for excitability increase in the ACC under PINP.
2) In his version of the manuscript the discussion is largely descriptive and repetitive, it is necessary to enrich the discussion taking in to account the importance of changes observed in the context of PINP.

Additional comments

Minor:
1) Rewrite several lines (l.) in the manuscript, for example: l.36; l.39; l.67; l.71; l.143.
2) Review the use of abbreviations throughout the manuscript, define them in their first appearance and use them throughout the entire manuscript, for example: l.11, Ct; l.12, Nav; l.19, PINP, some of them included in the title of the manuscript.
3) In results section state a precise quantification for each of the subunits that show increase in expression, for example Nav1.2 increases X±x n-fold, or as a percentage.

Reviewer 2 ·

Basic reporting

The manuscript by Willias Masocha shows an increment in the expression of alfa and beta subunits of different voltage gated sodium channels in the anterior cingular cortex (ACC) following to administration of the taxane paclitaxel, used as a chemotherapeutic agent.
Undoubtedly, since chemotherapy induced peripheral neuropathy is common, the results could have clinical relevance. However, there are a number of concerns about the study that should be addressed, most of them perhaps needs further experiments.

Experimental design

In my opinion, the main problem with the manuscript is the lack of evidence supporting the relationship between the increment in the expression levels of voltage gated sodium channels in the ACC and neuropathic pain.

The data of the mRNA expression should be represented in absolute, not relative units?
Why is the rationale of figure 2 and table 1?
Student t-test is not the best choice to compare relative units.
Is there any correlation between expression levels of voltage gated sodium channels and nociceptive behavior?

Validity of the findings

Additionally, the interpretation of the data regarding the relationship of the increment in the expression of the voltage gated sodium channels in the ACC and nociception produced by paclitaxel administration is confusing. The author did not shown any evidence that paclitaxel administration produces hyperalgesia or allodinia, so it is not possible to conclude that paclitaxel induced neuropathic pain. Moreover, paclitaxel may produce an increase of the expression of voltage gated sodium channels in other CNS structures related with pain processing or by other mechanisms (see Zhang et al, J Pain. 2016 doi: 10.1016/j.jpain.2016.02.011). These points should be discussed.

---

## Round 0.2 · Minor Revisions

· Academic Editor

Minor Revisions

Please follow the comments of reviewer #1 in a new revised version of the article.

Reviewer 1 ·

Basic reporting

The article meets the PeerJ standards.

Experimental design

Include expression values obtained from untreated control animals (i.e., without vehicle injection).

Validity of the findings

No comments

Additional comments

In the current version of the manuscript, the author has largely modified the text following the suggestions made in the first review. It is important to emphasize however that the information on protein expression is critical to subsequently define the meaning of expression changes in mRNA level.
In a new version untreated controls should be included.

Reviewer 2 ·

Basic reporting

The manuscript has been improved accordingly to my suggestions. I do not have more comments for the author

Experimental design

no more comments

Validity of the findings

no more comments

Additional comments

no more comments

---

## Round 0.3 · accepted · Accept

· Academic Editor

Accept

The revised version of the paper is now acceptable for publication.